# Soft Tissue Sarcoma Mimicking Melanoma: A Systematic Review

**DOI:** 10.3390/cancers15143584

**Published:** 2023-07-12

**Authors:** Fortunato Cassalia, Francesco Cavallin, Andrea Danese, Paolo Del Fiore, Claudia Di Prata, Marco Rastrelli, Anna Belloni Fortina, Simone Mocellin

**Affiliations:** 1Unit of Dermatology, Department of Medicine, University of Padua, 35121 Padua, Italy; fortunato1287@gmail.com; 2Independent Researcher, 36020 Solagna, Italy; cescocava@libero.it; 3Unit of Dermatology, Department of Integrated Medical and General Activity, University of Verona, 37100 Verona, Italy; adanese4@gmail.com; 4Soft-Tissue, Peritoneum and Melanoma Surgical Oncology Unit, Veneto Institute of Oncology IOV-IRCCS, 35128 Padua, Italy; marco.rastrelli@unipd.it (M.R.); simone.mocellin@unipd.it (S.M.); 5Department of Surgical, Oncological and Gastroenterological Sciences (DISCOG), University of Padua, 35128 Padua, Italy; claudia.diprata@gmail.com; 6Pediatric Dermatology Unit, Department of Medicine DIMED, University of Padua, 35121 Padua, Italy; anna.bellonifortina@unipd.it

**Keywords:** melanoma, sarcoma, mimicking

## Abstract

**Simple Summary:**

Sarcoma may show similarities to malignant melanoma, making it difficult to differentiate between these two neoplasms. This systematic review summarizes evidence on cases of sarcoma that were initially diagnosed as melanoma to help clinicians in the diagnostic process. A comprehensive search of key databases identified 23 case reports and 4 case series with a total of 34 patients. Heterogeneous clinical presentation and frequent immunohistochemistry positivity contributed to the initial misdiagnosis. The second assessment was performed due to unusual presentation or uncertain diagnosis, and the final diagnosis was clear cell sarcoma (50%) or other soft tissue sarcomas (50%). EWSR1 translocation was investigated in 50% of cases, among which 94% were found to be positive. This systematic review suggests that a second diagnosis should be considered in cases of atypical lesions, and ESWR1 translocation should be investigated.

**Abstract:**

Background: Sarcoma may show similarities to malignant melanoma in terms of morphologic and immunohistochemical aspects, making it difficult to differentiate between these two neoplasms during the diagnostic process. This systematic review aims to summarize available evidence on cases of sarcoma that were initially diagnosed as melanoma. Methods: A comprehensive search of the MEDLINE/Pubmed, EMBASE, and SCOPUS databases was conducted through March 2023. We included case series and case reports of sarcoma patients that were initially diagnosed as malignant melanoma. PRISMA guidelines were followed. Results: Twenty-three case reports and four case series with a total of 34 patients were included. The clinical presentation was heterogeneous, and the most involved anatomical regions were lower limbs (24%), head/neck (24%), and upper limbs (21%). IHC positivity was reported for S100 (69%), HMB45 (63%), MelanA (31%), and MiTF (3%). The main reasons for a second assessment were unusual presentation (48%) and uncertain diagnosis (28%). EWSR1 translocation was investigated in 17/34 patients (50%) and found to be positive in 16/17 (94%). The final diagnosis was clear cell sarcoma (50%) or other soft tissue sarcomas (50%). Conclusions: Melanoma and some histotypes of sarcoma share many similarities. In cases of atypical lesions, a second diagnosis should be considered, and ESWR1 translocation should be investigated.

## 1. Introduction

The 2020 World Health Organization (WHO) Classification of Soft Tissue Tumors indicates sarcomas as rare tumors that are further subclassified into approximately 70 subtypes, each characterized by a distinct morphology that often translates into a specific clinical behavior, as well as into specific therapeutic approaches [1,2]. They can occur anywhere in the body, affecting the extremities in 50% of cases, the trunk and retroperitoneum in 40% of cases, and the head and neck in 10% of cases [3].

Some of these neoplasms can be confused with cutaneous melanoma, since they share similar clinical, histological, and immunohistochemical features [4]. In addition, they are rare tumors that are often overlooked by clinicians who do not pose the diagnostic suspicion to the pathologist [2,5,6]. Therefore, healthcare specialists may encounter some difficulties in differentiating between sarcoma and melanoma during the diagnostic process. For example, clear cell sarcoma (CCS) clinically presents as a deep and small (<5 cm) soft tissue lesions, often juxtaposed with tendons, fascia, or aponeurosis, which may mistakenly suggest some forms of melanoma, such as acral melanoma, nodular melanoma, or amelanotic melanoma [7,8]. Moreover, CCS shows a phenotype identical to that of conventional melanoma characterized by strong expression of S100 protein in 100% of cases and variable expression of HMB-45, Melan A, and MiTF. As a matter of fact, CCS can be genetically differentiated from melanoma due to some peculiarities, including (i) the typical reciprocal translocation t(12;22) (q13;q12) that gives rise to the EWSR1-ATF1 oncogene and (ii) the absence of BRAF/NRAS mutations that can often characterize melanoma [4]. Other examples include malignant peripheral nerve sheath tumor (MPNST) and Kaposi’s sarcoma. MPNST usually arises from peripheral nerves and may be associated with patients with neurofibromatosis type 1 (NF1). The clinical presentation involves the development of a painful and/or rapidly expanding mass with associated neurological deficits. The biological behavior of MPNSTs has been described as unpredictable, and the differential diagnosis includes several tumors, particularly spindle cell/desmoplastic melanoma, which serves as the main differential because of its higher incidence, remarkably similar morphology, and overlapping immunochemical markers [9,10]. Kaposi’s sarcoma is typical of immunocompromised patients and may also present as a single papular skin lesion, clinically mimicking melanoma, which can be distinguished by histologic and immunohistochemical appearance [11]. 

The clinical presentation of such a neoplasm may mislead the physician, with potential implications for therapeutic strategy and prognosis. For example, surgical widening of the margins varies according to the neoplasm under treatment (melanoma or sarcoma); chemoradiotherapy may be offered for sarcoma, but melanoma patients may benefit from a different first-line approach, such as immunotherapy or target therapy or both, whereas sentinel lymph node biopsy is routinely performed in the diagnostic workup of melanoma but is still under debate for soft tissue sarcoma [12,13,14].

Awareness of the similarities between sarcoma and melanoma and the ability to recognize the two entities play a crucial role in patient care. However, a clinician may have little to no direct experience in this matter and may retrieve only limited information from a case report or a small case series.

The aim of this systematic review was to summarize available evidence on cases of sarcoma that were initially diagnosed as malignant melanoma to help clinicians in the diagnostic process and to improve patient care.

## 2. Materials and Methods

### 2.1. Study Design

This is a systematic review of case series and case reports describing cases of sarcoma that were initially diagnosed as malignant melanoma. The review was conducted according to the Preferred Reporting Items for Systematic Reviews and Meta-Analyses (PRISMA) guidelines [15]. The review protocol was registered in PROSPERO (CRD42023403882).

### 2.2. Search Strategy

We systematically searched MEDLINE/PubMed, EMBASE, and SCOPUS to detect eligible studies. The search strategy was conducted without language restrictions through March 2023. In PubMed, the following search strategy was used: sarcoma mimicking melanoma OR sarcoma resembling melanoma OR melanoma-like. The search strategy was tailored to conform to the other electronic sources. The lists from each source were joined, and the duplicates were removed. Two investigators (F.C. and A.D.) separately evaluated titles and abstracts of the records and removed those that fell outside the scope of the review. The full texts of all potentially eligible records were examined to dismiss those not fulfilling the inclusion criteria. Finally, the reference lists of included records were hand-searched to detect further studies of interest. Any disagreement was solved by consensus with a third investigator (P.D.F.). Studies not including human subjects were excluded. No language restrictions were applied.

### 2.3. Data Collection

Two investigators (F.C. and A.D.) independently extracted relevant data from the included articles. For each article, study features, patient characteristics, tumor information, and outcome measures were collected. A third investigator (A.B.F.) checked the extracted data. Any inconsistency was solved by consensus.

### 2.4. Assessment of the Quality of Included Studies

The quality of the included studies was assessed according to eight criteria: (i) clear criteria for inclusion of the patient(s); (ii) valid methods for identification of the initial condition; (iii) valid methods for identification of the final condition; (iv) in a case series, consecutive inclusion of patients; (v) clear reporting of demographics; (vi) clear reporting of clinical information; (vii) reporting of the time of the second assessment; and (viii) reporting of the reason for the second assessment. The criteria were adapted from the Joanna Briggs Institute (JBI) critical appraisal tool [16] to fit the context under evaluation (case series and case reports describing cases of sarcoma that were initially diagnosed as malignant melanoma). Two investigators (F.C. and M.A.) independently appraised the risk of bias of the included studies, and any inconsistency was solved by consensus with all authors.

### 2.5. Data Synthesis

The selection procedure was presented in a flow chart. Pertinent data were extracted from included studies and summarized in tables. The inclusion of case reports and very small case series precluded the feasibility of a meaningful meta-analysis; hence, a narrative synthesis of included studies was conducted.

## 3. Results

### 3.1. Search Results

The comprehensive search of key databases yielded 365 non-duplicate records. We excluded 321 records according to the title or the abstract, and we identified 43 potentially eligible records for the full-text review. During this phase, 24 records satisfied the inclusion criteria, while 19 records were excluded due to different design (*n* = 4), different topic (*n* = 12), or different participants (*n* = 1) or because we could not find the full text (*n* = 2) (Appendix A). Three additional eligible records were identified via hand search. Finally, 27 records [9,11,17,18,19,20,21,22,23,24,25,26,27,28,29,30,31,32,33,34,35,36,37,38,39,40,41] were included in the narrative synthesis (Figure 1).

### 3.2. Narrative Synthesis of the Findings

The synthesis included 23 case reports (85%) and 4 case series (15%). A total of 3 studies (11%) were published in 1989–2000, 7 (26%) in 2001–2011, and 17 (63%) in 2012–2022. Study and patient characteristics are summarized in Table 1 and Table 2. 

Overall, the studies reported on a total of 34 patients (aged 12–86 years), including 25 males (74%) and 9 females (26%). The initial diagnosis was melanoma (24 patients, 71%) or suspected melanoma (10 patients, 29%). The clinical presentation was heterogeneous (Table 1), and the involved anatomical regions included lower limbs (8/33 patients, 24%), head/neck (8/33 patients, 24%), upper limbs (7/33 patients, 21%), visceral area (4/33 patients, 12%), trunk (3/33 patients, 9%), and genital area (3/33 patients, 9%) (the information was not available for one patient).

IHC positivity was reported for S100 (22/32 patients, 69%), HMB45 (20/32 patients, 63%), MelanA (10/32 patients, 31%), and MiTF (1/32 patients, 3%).

The reasons for a second assessment and/or diagnostic re-evaluation included unusual presentation (12/25, 48%), uncertain diagnosis (7/25, 28%), expert opinion (2/25, 8%), no response to treatment (1/25, 4%), search for EWSR1 translocation (1/25, 4%), review after surgery (1/25, 4%), and review after death (1/25, 4%), while the information was not reported in nine patients.

EWSR1 translocation was investigated in 17/34 patients (50%) and found to be positive in 16 CCS patients and 1 MPNST patient.

The final diagnosis was clear cell sarcoma in 17 patients (50%) and soft tissue sarcoma in 17 patients (50%). The latter included sarcoma of perivascular epithelioid cells (*n* = 4), malignant peripheral nerve sheath tumor (*n* = 4), Kaposi’s sarcoma (*n* = 1), chondroid syringoma (*n* = 1), cutaneous angiosarcoma (*n* = 1), cutaneous epithelioid angiosarcoma (*n* = 1), epithelioid malignant schwannoma (*n* = 1), malignant giant cell tumor of soft tissue (*n* = 1), malignant schwannoma (*n* = 1), myeloid sarcoma (*n* = 1), and pleomorphic sarcoma (*n* = 1).

An overview of the main findings is displayed in Figure 2.

### 3.3. Critical Appraisal of the Quality of Included Studies

Table 3 summarizes the quality assessment of the included studies. All studies (27/27, 100%) reported clear criteria for inclusion of the patient(s). Valid methods for identification of the initial (melanoma) and final (sarcoma) conditions were described by 16/27 (59%) and 18/27 (67%) studies, respectively. All case series (4/4, 100%) included consecutive patients. Clear reporting of demographics and clinical information were found in 27/27 (100%) and 24/27 (89%) studies, respectively. Only one study (4%) specified the timing of the second assessment, and 17/27 studies (63%) clearly reported the reason for the second assessment.

## 4. Discussion

This systematic review evaluated the available evidence on cases of sarcoma that were initially diagnosed as malignant melanoma. Our search yielded only case reports and small case series [9,11,17,18,19,20,21,22,23,24,25,26,27,28,29,30,31,32,33,34,35,36,37,38,39,40,41], which, individually, can provide sparce information to healthcare providers; however, summarizing data from such sources may allow for a better understanding of the topic. Most of the studies were published in the last decade, which may suggest a rising interest in differentiating sarcoma from melanoma during the diagnostic process. 

Overall, the clinical presentation of such cases was heterogeneous, and some sarcomas were initially misdiagnosed because several aspects, such as clinical factors, localization of the lesion, and histologic appearance, suggested a malignant melanoma [9,17,18,19,20,21,22,23,24,25,26,27,28,32,33,34,35,39,40,41]. In some cases, the clinician considered other diagnoses but finally opted for melanoma [29,30,31,36,37,38]. Furthermore, immunohistochemistry is not helpful for differentiating sarcoma from melanoma, which was suggested by the positivity of some markers, such as S100, HBMG-45, and MelanA [9,17,18,19,20,21,22,23,26,27,29,30,31,32,33,34,35,36,37,38,39]. Hence, the rarity of cases of sarcoma mimicking melanoma likely played an important role in opting for melanoma as the reasonable initial diagnosis of choice. The reader should be aware that we assume such rarity given the few cases in the literature, but we do not have robust information about the real magnitude of cases of sarcoma mimicking melanoma. 

In fact, the main reasons for the second assessment leading to a diagnosis of sarcoma were unusual presentation and uncertainty about the diagnosis, which suggested further investigations to the clinicians [9,18,19,20,21,22,23,29,30,32,33,35,38]. In a few cases, the second assessment was performed because the patient did not respond to the treatment [17] or during a retrospective review of cases [21,24]. 

This implies that the correct identification of a sarcoma mimicking melanoma relies on the healthcare provider being aware of the possibility of such a case and being able to identify when unusual features merit further investigation. 

This also means that the prevalence of such cases is unknown because the literature does not include episodes when the healthcare provider did not feel the need for further investigations, and no systematic investigations have been conducted in large series of melanoma patients.

We believe that misdiagnosing a sarcoma as a melanoma may have potential implications for patient care because of the use of different therapeutic approaches, including sentinel node biopsy, first-line therapy, and surgical therapy [12,13,14]. Unfortunately, available information is insufficient to assess the prognostic effect of such misdiagnosis.

Interestingly, half of the sarcomas found at the second assessment were CCS [17,18,19,20,21,22,23,26,29,30,31,32,33]. We believe that the common features shared by CCS and melanoma [7,8] and the lower incidence of CCS were likely to be responsible for the initial misdiagnosis. When investigated, EWSR1 translocation was found to be positive in almost all cases [17,18,19,20,21,22,26,29,30,31,32,33]; hence, clinicians may benefit from the investigation of EWSR1 translocation in the initial diagnostic process. 

This systematic review has some limitations that should be considered by the reader. First, the research topic was prone to be described in case reports and very small case series, limiting the available information and the potential for further analyses. Second, the lack of epidemiological studies prevented any considerations of the prevalence of cases of sarcoma that were initially diagnosed as melanoma. Third, information about the timing of the second assessment would provide interesting information but was largely missing in the literature. Fourth, the role of EWSR1 translocation in the identification of CCS could not be investigated because of selected reporting (the included studies reported some CCS cases with positive EWSR1 and one MPNST cases with negative EWSR1).

Within its limitations, our systematic review underlines an underreported problem in the diagnosis of melanoma and sarcoma, informs physicians about features that can make differential diagnosis difficult, and highlights the importance of searching for EWSR1 translocation in the diagnostic process. Due to the rarity of sarcoma, healthcare providers possess a heterogeneous level of experience and expertise in managing such diseases. Therefore, it is crucial for physicians to ensure that pathologists are appropriately guided to relevant diagnostic procedures, especially when excising suspicious melanoma lesions in centers without specialized knowledge of sarcoma. The histologic and immunohistochemical similarities between melanoma and sarcoma can occasionally present a challenge for less experienced pathologists, increasing the risk of misdiagnosis. Therefore, effective collaboration among physicians, surgeons, and pathologists is essential to accurately guide the diagnostic process and assist pathologists in reaching a definitive histologic diagnosis. Physicians should consider sarcoma—particularly CCS, as mentioned—as a plausible differential diagnosis when encountering lesions that lack the typical clinical features of melanoma, especially those located deep within or near tendons and/or aponeurosis, particularly in young patients. It is noteworthy that the definitive diagnosis of CCS often relies on identifying EWSR1 translocation. Therefore, physicians should provide explicit guidance to pathologists, enabling them, when necessary, to actively search for EWSR1 translocation to definitively confirm the diagnosis. Alternatively, in the absence of clear guidance from the physician, a less experienced pathologist facing difficulties in reaching a definitive diagnosis for a suspected melanoma lesion should seek a second opinion from more experienced colleagues. This proactive approach may facilitate the timely implementation of appropriate therapeutic interventions, ultimately leading to improved patient outcomes and, potentially, prognosis.

## 5. Conclusions

Atypical skin lesions may be misdiagnosed as melanomas if they share many similarities. Physicians should be aware of such a possibility in the diagnostic process, as it may have potential implications for the treatment strategy. In the case of atypical skin lesions, it may be useful to investigate the presence of EWSR1 translocation, since CSS are the most common histology to be found in case of re-evaluation. Referral to tertiary expert centers may be recommended. Further investigations are required to better understand the epidemiology of misleading diagnosis and to raise awareness of the issue.

## Figures and Tables

**Figure 1 cancers-15-03584-f001:**
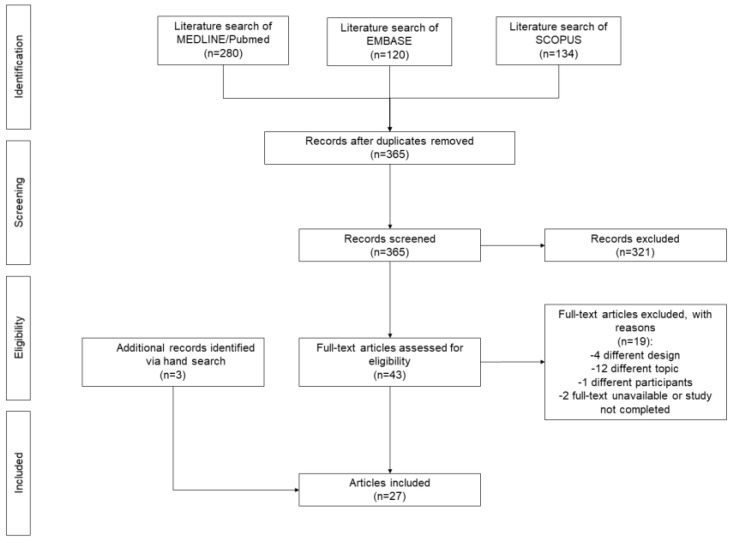
Flow chart of the selection process.

**Figure 2 cancers-15-03584-f002:**
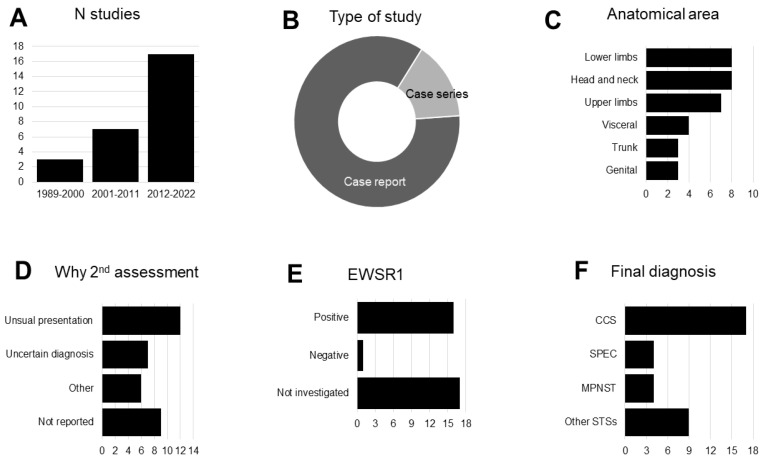
Overview of main findings. CCS: clear cell sarcoma; MPNST: malignant peripheral nerve sheath tumor; SPEC: sarcoma of perivascular epithelioid cells; STS: soft tissue sarcoma.

**Table 1 cancers-15-03584-t001:** Characteristics of included studies: patient characteristics and clinical presentation.

First Author	Year	Type of Study	N pts	Sex	Age (Years)	Initial Diagnosis	Anatomical Region	Site	Clinical Presentation	IHC Positivity
Potter AJ [17]	2022	Case report	1	M	30	Melanoma	Lower limbs	Toe	Ulcerated, nodular cutaneous lesion on the distal third toe, which had been present for several years	S100, HMB45, SOX10
Tahiri EL [18]	2022	Case report	1	M	31	Melanoma	Lower limbs	Heel	Heel mass nodule	S100, HMB45, Melan A
Biglow LR [9]	2021	Case report	1	F	47	Melanoma	Upper limbs	Finger	Subcutaneous nodule at the finger without any obvious nevus or skin color changes	S100, SOX10, Vimentin, BCL2
Zhang X [19]	2021	Case report	1	M	68	Melanoma	Visceral	Pleura	Dyspnea and cough following a dental abscess that was treated with root canal procedure; imaging studies revealed a large right pleural effusion, raising the concern of an empyema	SOX10, S100, HMB45, Melan-A
Nawrocki S [20]	2020	Case report	1	M	25	Melanoma	Lower limbs	Left inguinal region	Raised blood blister that changed colors	S100, HMB45, Melan A
Obiorah IE [21]	2019	Case series	2	F	37	Melanoma	Head and neck	Left neck	Complaint of left jaw pain and swelling	HMB45, S100, CD31, CD34, CD68
				M	33	Melanoma	Trunk	Back	Mid-back pain radiating to the flanks, as well as leg weakness and numbness, with gait abnormalities	HMB45, S100, Vimentin
Donzel M [22]	2019	Case report	1	M	27	Melanoma	Head and neck	Palate	Palatal ulcerations; ill-defined and erythematous, with a friable center, superficial erosions, and irregular, raised edges	HMB45, SOX10, Melan A
Obiorah IE [23]	2018	Case report	1	F	43	Melanoma	Head and neck	Right neck	Small nodule on the right side of her neck	S100, HMB45, Vimentin
Curry JL [24]	2018	Case report	1	M	68	Melanoma	Upper limbs	Left upper harm	Primary tumor not told; recurrence: new; slightly tender; 1.0 cm purpuric cutaneous nodule within the lymphatic drainage field of his previous primary melanoma of his left upper arm	CD11, CD43, CD68
Leon-Castillo A [25]	2017	Case report	1	M	65	Melanoma	Head and neck	Occipital scalp	Large multinodular, cutaneous occipital scalp lesion with erythematous background	CD31, ERG, D2-40, factor VIII–related antigen, Tyrosinase, HMB45, Melan A
Zivanovic M [26]	2017	Case report	1	M	20	Melanoma	Lower limbs	Foot	N/A	S100, Melan A, HMB45
Jackson CR [27]	2016	Case report	1	M	56	Melanoma	Trunk	Chest	Flesh-colored chest lesion for 7 years	S100, SOX10, CD34
Castriconi M [28]	2015	Case report	1	M	56	Melanoma	Upper limbs	Right axilla	Giant mass located on the right axilla	N/A
Sayah M [29]	2015	Case report	1	F	54	Suspect of melanoma	Visceral	Cecum	Severe iron deficiency anemia and hematochezia	S100, Cytokeratins, HMB45
Liu C [30]	2014	Case series	2	M	29	Suspect of melanoma	Upper limbs	Left thumb	Solid gray–white tumor	HMB45, Melan A, CD56, S100, Vimentin, NSE
				M	76	Suspect of melanoma	Visceral	Jejunum	Complaining of bowel obstruction, macroscopic examination: tumor (2.5 cm × 2.2 cm × 1.5 cm) with a whitish–gray surface	S100, Vimentin, NSE
Sidiropoulos M [31]	2013	Case report	1	M	13	Suspect of melanoma	Head and neck	Lower lip	Symptomatic papule on the lower lip that was suggestive of a mucocele	S100, CD99, sinaptofisina, HMB45, MiTF
Falconieri G [32]	2012	Case series	3	M	12	Melanoma	Lower limbs	Left foot	Lesion in the dorsal aspect of foot	S100, Melan A
				M	60	Melanoma	Lower limbs	Upper thigh	Slowly growing pigmented nodular lesion	S100
				F	29	Melanoma	Lower limbs	Right foot	Lesion in the sole of the foot	S100, Melan A, HMB45
Rodríguez MM [33]	2009	Case report	1	M	53	Melanoma	Upper limbs	Right harm	Painful erythematous, dome-shaped, nodular lesion 1.3 cm in diameter, firm to palpation and movable, with a serohemorrhagic crust on its surface	S100, HMB45
Tanas MR [34]	2009	Case report	1	M	67	Melanoma	Trunk	Abdomen	Abdominal mass	S100, HMB45, MiFT, Melan A
Zoufaly A [11]	2007	Case report	1	M	69	Melanoma	N/A	N/A	N/A	N/A
Brightman LA [35]	2006	Case report	1	M	86	Melanoma	Head and neck	Scalp	Large irregular dark gray–blue plaque with an adjacent speckled tan nodule	S100, CD31, CD34
Matsuda Y [36]	2005	Case report	1	M	75	Suspect of melanoma	Lower limbs	Left thigh	Oval-shaped mass; elastic, soft, and adherent to the left thigh on palpation	S-100, NSE, GFAP, MBP, Chromogranin A and synaptophysin
Demir Y [37]	2003	Case report	1	M	80	Suspect of melanoma	Head and neck	Scalp	Painless ulceration on his scalp	S100
Bonetti F [38]	2001	Case series	4	F	28	Suspect of melanoma	Visceral	Ileum	Abdominal pain	HMB45, MART 1
				F	19	Suspect of melanoma	Genital	Uterus	Abdominal pain	HMB45
				F	40	Suspect of melanoma	Genital	Uterus	Surgery because of uterine leiomyomas; during the operation, a 2.5 cm × 12 cm × 1.5 cm pelvic nodule was accidentally found and thought to represent endometriosis	HMB45
				F	41	Suspect of melanoma	Genital	Myometrium	Presumed fibroids in uterus	HMB45, MART 1
Ferreiro JA [39]	1995	Case report	1	M	75	Melanoma	Head and neck	Face	Non-painful mass of the face	keratin, S100, Vimentin
Honma K [40]	1989	Case report	1	M	65	Melanoma	Upper limbs	Left axillary fossa	N/A	Leu7, NSE
Gould E [41]	1989	Case report	1	M	78	Melanoma	Upper limbs	Left arm	Black nodule above the elbow	a1 anti-chymotrypsin (AACT), a1 antitrypsin (AAC)

**Table 2 cancers-15-03584-t002:** Characteristics of included studies: second assessment and final diagnosis.

First Author	Year	Why Second Assessment and/or Diagnostic Re-Evaluation	EWSR1 Translocation	Final Diagnosis
Potter AJ [17]	2022	Unresponsive to treatment	Positive	Clear cell sarcoma
Tahiri EL [18]	2022	Unusual presentation	Positive	Clear cell sarcoma
Biglow LR [9]	2021	Unusual presentation	Negative	Malignant peripheral nerve sheath tumor
Zhang X [19]	2021	Uncertain diagnosis	Positive	Clear cell sarcoma
Nawrocki S [20]	2020	Uncertain diagnosis	Positive	Clear cell sarcoma
Obiorah IE [21]	2019	Review after death	Positive	Clear cell sarcoma
		Uncertain diagnosis	Positive	Clear cell sarcoma
Donzel M [22]	2019	Uncertain diagnosis	Positive	Clear cell sarcoma
Obiorah IE [23]	2018	Uncertain diagnosis	N/A	Clear cell sarcoma
Curry JL [24]	2018	Review after surgery	N/A	Myeloid sarcoma
Leon-Castillo A [25]	2017	N/A	N/A	Cutaneous angiosarcoma
Zivanovic M [26]	2017	N/A	Positive	Clear cell sarcoma
Jackson CR [27]	2016	Expert opinion	N/A	MPNST—malignant peripheral nerve sheath tumors
Castriconi M [28]	2015	N/A	N/A	Pleomorphic sarcoma
Sayah M [29]	2015	Unusual presentation	Positive	Clear cell sarcoma
Liu C [30]	2014	Unusual presentation	Positive	Clear cell sarcoma
		Unusual presentation	Positive	Clear cell sarcoma
Sidiropoulos M [31]	2012	Research for EWSR1 translocation	Positive	Clear cell sarcoma
Falconieri G [32]	2012	Unusual presentation	Positive	Clear cell sarcoma
		Unusual presentation	Positive	Clear cell sarcoma
		Unusual presentation	Positive	Clear cell sarcoma
Rodríguez MM [33]	2009	Uncertain diagnosis	Positive	Clear cell sarcoma
Tanas MR [34]	2009	N/A	N/A	Malignant peripheral nerve sheath tumor
Zoufaly A [11]	2007	N/A	N/A	Kaposi’s sarcoma
Brightman LA [35]	2006	Uncertain diagnosis	N/A	Cutaneous epithelioid angiosarcoma
Matsuda Y [36]	2005	N/A	N/A	Malignant peripheral nerve sheath tumor
Demir Y [37]	2003	Expert opinion	N/A	Malignant schwannoma
Bonetti F [38]	2001	Unusual presentation	N/A	Sarcoma of perivascular epithelioid cells
		Unusual presentation	N/A	Sarcoma of perivascular epithelioid cells
		Unusual presentation	N/A	Sarcoma of perivascular epithelioid cells
		Unusual presentation	N/A	Sarcoma of perivascular epithelioid cells
Ferreiro JA [39]	1995	N/A	N/A	Chondroid syringoma
Honma K [40]	1989	N/A	N/A	Epithlioid malignant schwannoma
Gould E [41]	1989	N/A	N/A	Malignant giant cell tumor of soft tissue

**Table 3 cancers-15-03584-t003:** Summary of the quality assessment of the included studies.

First Author	Year	Clear Criteria for Inclusion	Valid Methods for the Identification of the Initial Condition	Valid Methods For Identification of the Final Condition	In a Case Series, Consecutive Inclusion of Participants	Clear Reporting of Demographics	Clear Reporting of Clinical Information	Reporting of Time of Second Assessment	Reporting of Reason for Second Assessment
Potter AJ [17]	2022	Yes	Yes	Yes	N/A	Yes	Yes	No	Yes
Tahiri EL [18]	2022	Yes	Yes	yes	N/A	Yes	Yes	No	Yes
Biglow LR [9]	2021	Yes	Yes	Unclear	N/A	Yes	Yes	No	Yes
Zhang X [19]	2021	Yes	Yes	Yes	N/A	Yes	Yes	Unclear	Yes
Nawrocki S [20]	2020	Yes	Yes	Yes	N/A	Yes	Yes	No	Yes
Obiorah IE [21]	2019	Yes	Yes	Yes	Yes	Yes	Yes	Yes	Yes
Donzel M [22]	2019	Yes	Yes	Yes	N/A	Yes	Yes	No	Yes
Obiorah IE [23]	2018	Yes	Yes	Yes	N/A	Yes	Yes	No	Yes
Curry JL [24]	2018	Yes	Yes	Yes	N/A	Yes	Yes	No	Yes
Leon-Castillo A [25]	2017	Yes	Yes	Yes	N/A	Yes	Yes	No	No
Zivanovic M [26]	2017	Yes	Yes	Yes	N/A	Yes	No	No	No
Jackson CR [27]	2016	Yes	Yes	Unclear	N/A	Yes	Yes	No	Yes
Castriconi M [28]	2015	Yes	Yes	Unclear	N/A	Yes	Yes	No	No
Sayah M [29]	2015	Yes	Unclear	Yes	N/A	Yes	Yes	No	Yes
Liu C [30]	2014	Yes	Unclear	Yes	Yes	Yes	Yes	No	Yes
Sidiropoulos M [31]	2012	Yes	Unclear	Yes	N/A	Yes	Yes	No	Yes
Falconieri G [32]	2012	Yes	Yes	Yes	Yes	Yes	Yes	No	Yes
Rodríguez MM [33]	2009	Yes	Unclear	Yes	N/A	Yes	Yes	No	Unclear
Zoufaly A [11]	2007	Yes	Yes	No	N/A	Yes	No	No	No
Tanas MR [34]	2009	Yes	Unclear	Unclear	N/A	Yes	Yes	No	No
Brightman LA [35]	2006	Yes	Yes	Yes	N/A	Yes	Yes	No	Yes
Matsuda Y [36]	2005	Yes	Unclear	Unclear	N/A	Yes	Yes	No	No
Demir Y [37]	2003	Yes	Unclear	Yes	N/A	Yes	Yes	No	Yes
Bonetti F [38]	2001	Yes	Unclear	Yes	Yes	Yes	Yes	No	Yes
Ferreiro JA [39]	1995	Yes	Unclear	Unclear	N/A	Yes	Yes	No	No
Honma K [40]	1989	Yes	Unclear	Unclear	N/A	Yes	No	No	No
Gould E [41]	1989	Yes	Unclear	unclear	N/A	Yes	Yes	No	No

## Data Availability

The data presented in this study are available within the article.

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
