# Peer review of "Soft Tissue Sarcoma Mimicking Melanoma: A Systematic Review"

_cancers, 2023, doi:10.3390/cancers15143584_

Round 1
Reviewer 1 Report
The authors aimed to summarize the available evidence on cases of sarcoma that were initially diagnosed as malignant melanoma, to help clinicians in the diagnostic process, to improve patient care.
The topic is interesting and may fit better a pathology journal.
The manuscript is a well conducted systematic review. I have a few comments to the authors to improve the clarity of their manuscript.
1- The authors excluded 17 articles. The authors are advised to list all of the excluded studies in a table outlining the reason for exclusion.
2- The discussion is under-developed and can benefit from providing a guide to help pathologists in making the correct diagnoses. What are the precautionary measures that have to be implemented to prevent any misdiagnosis.
The Manuscript is well written.
Author Response
Reviewer #1
- The authors excluded 17 articles. The authors are advised to list all of the excluded studies in a table outlining the reason for exclusion.
Re: We provided such information in the Supplementary Table 1 of the revised manuscript.
Point 2: The discussion is under-developed and can benefit from providing a guide to help pathologists in making the correct diagnoses. What are the precautionary measures that have to be implemented to prevent any misdiagnosis.
Re: We acknowledge that the Discussion section had some limitations, because of the sparse information in the literature which suggests caution in the interpretation of the findings. We thank the Reviewer for the comment and we included some considerations about possible indications for a pathologist and the precautionary measures to prevent a misdiagnosis: “Due to the rarity of sarcoma, healthcare providers possess a heterogeneous level of experience and expertise in managing such disease. Therefore, it is crucial for physicians to ensure that pathologists are appropriately guided to relevant diagnostic procedures, especially when excising suspicious melanoma lesions in centers without specialized knowledge of sarcoma. The histologic and immunohistochemical similarities between melanoma and sarcoma can occasionally present a challenge for less experienced pathologists, increasing the risk of misdiagnosis. Therefore, effective collaboration among physicians, surgeons, and pathologists is essential to accurately guide the diagnostic process and assist pathologists in reaching a definitive histologic diagnosis. Physicians should consider sarcoma, particularly CCS as mentioned, as a plausible differential diagnosis when encountering lesions that lack the typical clinical features of melanoma, especially those located deep within or near tendons and/or aponeurosis, particularly in young patients. It is noteworthy that the definitive diagnosis of CCS often relies on identifying the EWSR1 translocation. Therefore, physicians should provide explicit guidance to pathologists, enabling them, when necessary, to actively search for the EWSR1 translocation to definitively confirm the diagnosis. Alternatively, in the absence of clear guidance from the physician, a less experienced pathologist facing difficulties in reaching a definitive diagnosis for a suspected melanoma lesion should seek a second opinion from more experienced colleagues. This proactive approach may facilitate the timely implementation of appropriate therapeutic interventions, ultimately leading to improved patient outcomes and potentially prognosis.” (Discussion secton, page 11).

Reviewer 2 Report
1. Review article doesn't need to have these parts. Materials and Methods & Results
2. Please rewrite the results "The search identified 365 non‑duplicated records. After excluding 321 records based on title/abstract, 43 potentially eligible records were retrieved for full text review. Of these, 17 were excluded due to different design (n=4), different topic (n=12) or different participants (n=1). Another two records were excluded because the full‑text could not be found. Three additional records were identified via hand search. Finally, 27 records [9,11,17‑41] were included in the narrative synthesis (Figure 1).
3. Please recategorize the review article once again.
Moderate editing of English language is required.
Author Response
Reviewer #2
- Review article doesn't need to have these parts. Materials and Methods & Results.
Re: This is a systematic review and we believe that the organization in the usual sections (Introduction, Materials and Methods, Results, Discussion) improves readability and clarity for the reader. The journal requirements do not exclude such arrangment of the text in a paper presenting a systematic review and recent systematic reviews have been published in this journal with the text organized in the usual sections (Introduction, Materials and Methods, Results, Discussion). Therefore, we would like to keep the current format.
- Please rewrite the results "The search identified 365 non‑duplicated records. After excluding 321 records based on title/abstract, 43 potentially eligible records were retrieved for full text review. Of these, 17 were excluded due to different design (n=4), different topic (n=12) or different participants (n=1). Another two records were excluded because the full‑text could not be found. Three additional records were identified via hand search. Finally, 27 records [9,11,17‑41] were included in the narrative synthesis (Figure 1).
Re: We rewrote that section in the revised manuscript: “The comprehensive search of key databases yielded 365 non-duplicate records. We excluded 321 records according to the title or the abstract, and we identified 43 potentially eligible records for the full-text review. During this phase, 24 records satisfied the inclu-sion criteria while 19 records were excluded due to different design (n=4), different topic (n=12), different participants (n=1) or because we could not find the full-text (n=2) (Supplementary Table 1). Three additional eligible records were identified via hand-search. Finally, 27 records [9,11,17-41] were included in the narrative synthesis (Figure 1).” (page 3).
Point 3: Please recategorize the review article once again.
Re: We reviewed the metadata accordingly.

Round 2
Reviewer 2 Report
The article can be published without further revision.